# Comparison of the Outcomes of Double-Bar and Non-Double-Bar Techniques in the Minimally Invasive Repair of Pectus Excavatum: A Focus on Bar Removal and Complications

**DOI:** 10.3390/jcm14124217

**Published:** 2025-06-13

**Authors:** Duk Hwan Moon, Doyun Heo, Sungsoo Lee

**Affiliations:** 1Department of Thoracic and Cardiovascular Surgery, Gangnam Severance Hospital, Yonsei University College of Medicine, Seoul 06273, Republic of Korea; pupupuck@yuhs.ac; 2College of Medicine, Hanyang University Hospital, Seoul 04763, Republic of Korea; doyun9975@gmail.com

**Keywords:** pectus excavatum, MIRPE, double-bar technique, bar removal, CXR bony spur

## Abstract

**Background/Objectives:** Minimally invasive repair of pectus excavatum (MIRPE) techniques are classified into non-double-bar methods, which elevate the chest wall only, and double-bar methods, which involve both elevation and compression. Our objective was to compare the clinical outcomes of MIRPE procedures between the double-bar and non-double-bar techniques, with a particular focus on bar removal operation time and its associated complications. **Methods**: This retrospective study included 103 patients who were diagnosed with pectus excavatum and underwent MIRPE between January 2015 and July 2022. Propensity score matching was performed to adjust for confounding variables between patients who underwent the double-bar procedure and those who underwent the non-double-bar procedure. Comparative analyses between the groups were conducted using t-tests and chi-square tests, and linear regression was used to identify the risk factors for a prolonged bar removal time. **Results**: After propensity score matching, 48 patients were matched, with each group consisting of 24 patients. The double-bar group had a shorter hospital stay after bar insertion (*p* < 0.001), a reduced bar removal operation time (*p* < 0.001), and a lower incidence of chest X-ray bony spurs (*p* = 0.018) compared to the non-double-bar group. Multivariate linear regression model showed non-double-bar use and the presence of a chest X-ray bony spur were significant risk factors for a prolonged bar removal operation time. **Conclusions**: The double-bar technique demonstrated significant advantages, such as reduced bar removal operation times and fewer complications, making it the preferable surgical approach. Nevertheless, further prospective studies with larger sample sizes are needed to validate these findings.

## 1. Introduction

Pectus excavatum (PE) is a congenital deformity of the chest wall characterized by posterior displacement of the sternum and costal cartilages [1]. Patients with PE often grapple with psychological issues, such as low self-esteem and body image dissatisfaction [2]. Additionally, these patients may experience dyspnea during exertion, non-specific chest pain, and reduced exercise tolerance, with the possibility of experiencing pulmonary complications [3,4].

Since the advent of conventional minimally invasive repair of pectus excavatum (MIRPE), surgical methods for treating PE have advanced significantly [5]. These advancements have led to an increased surgical correction rate; a reduction in complications, including pectus bar dislocation; and improved patient prognoses [6,7]. Initially, methods involving the use of two or more bars, such as the cross-bar and parallel bar techniques, were introduced to enhance the stability and efficiency of the pectus bar [8]. However, in non-double-bar techniques, sternal elevation alone is often insufficient to achieve complete correction of asymmetric PE due to the limited distribution of corrective force [9]. Moreover, excessive mechanical stress at hinge points can lead to the formation of bony spurs around the bar, making bar removal technically difficult [10].

Recent suggestions indicate that the double-bar technique, which involves the insertion of two bars—one into the retrosternal region and one into the presternal region, with the aim of simultaneously elevating and compressing the chest wall—may significantly reduce operation times, the postoperative hospital admission period, and complication rates for PE patients [11]. The double-bar technique is continually being refined and improved, while other methods, such as the oblique double-bar technique, are being developed [9].

Despite these advancements, there is limited evidence directly comparing the clinical outcomes of the double-bar (DB) and non-double-bar (NDB) techniques, particularly in terms of the operation time for bar removal and associated complications. This study aims to compare the clinical outcomes between DB and NDB patients, with a specific focus on bar removal operation time, complications, and the incidence of chest X-ray bony spurs. By evaluating these facets, our goal is to provide insights into which technique offers superior outcomes, leading to better patient management and surgical decision-making in PE treatment.

## 2. Materials and Methods

### 2.1. Patients

Among patients with past medical records from the thoracic surgery department of a tertiary university hospital, 103 patients who were diagnosed with PE and had undergone an MIRPE operation between January 2015 and July 2022 were retrospectively selected. In this study, chest X-ray (CXR) bony spurs are defined as the presence of small, bilateral, extra pieces of bone that have grown on the surface of the rib bones, as observed on chest X-ray following bar removal operation. This study received approval from the local institutional review board of Gangnam severance Hospital in Korea (IRB No. 3-2025-0034), and the requirement for informed consent was waived due to its retrospective nature.

### 2.2. Operative Techniques

For patients in the DB group, the double-bar correction technique was employed to stabilize the chest wall. First, a primary bar was inserted below the chest wall along a path created by a passer, which was guided through chest tubes and then rotated into position [9]. Next, a second bar was placed subcutaneously and submuscularly above the chest wall using a second passer [9]. Once both bars were properly aligned, they were fixed together with two screws, ensuring stability during the postoperative period [9]. The configuration of the DB varied based on the patient’s chest shape, ranging from a single horizontal placement to a single oblique one, with some cases involving double-bar insertion at the lower part and a parallel single bar connection at the upper part or even double-bar insertion at both the upper and lower parts to create a parallel structure (Figure 1).

For patients in the NDB group, two bar correction techniques were employed: the cross type and the parallel type. In the case of the cross type, the crossing point of the two bars elevated both the deepest portion of the lesion and the entire chest wall [8]. In contrast, the parallel type involved placing the first bar transversely at the deepest point of the chest and the second bar transversely 1 to 2 intercostal spaces above the first [8]. In both techniques, the ends of the bar were secured using bridge plates [8].

### 2.3. Statistical Analysis

Descriptive statistics were used to summarize the following variables: age, gender, height, weight, body mass index (BMI), Haller index (HI), bar insertion operation variables, and bar removal operation variables. Independent sample *t*-tests were used to assess the differences between groups, and chi-square tests were used for categorical variables. Univariate and multivariate regression analyses were used to identify risk factors leading to prolonged removal operation time.

To adjust for confounding factors, propensity score matching was employed. Six confounding variables (age, gender, height, weight, BMI, and Haller index) were used to estimate the propensity score, which is the predicted probability of undergoing DB insertion. A logistic regression model was fitted to these six variables to predict the probability of undergoing DB insertion versus NDB insertion. Patients were then matched using the nearest-neighbor matching algorithm with a caliper of 0.2 standard deviations of the logit of the propensity scores. Each DB patient was matched 1:1 with an NDB patient with a similar propensity score.

All statistical analyses were performed using R software (version 4.4.0; R Foundation for Statistical Computing, Vienna, Austria).

## 3. Results

### 3.1. Before Propensity Score Matching

A total of 103 PE patients were divided into two groups: a double-bar (DB) group (n = 59) and a non-double-bar (NDB) group (n = 44). In terms of baseline characteristics, the DB group had significantly lower mean values for age, weight, height, and BMI (*p* < 0.001) compared to the NDB group. Additionally, the Haller index (HI) was significantly lower in the DB group than in the NDB group (*p* = 0.045). Among the variables related to bar insertion, hospital stay duration was significantly shorter (*p* < 0.001), and the complication rate was lower (*p* = 0.012), in the DB group compared to the NDB group. Bar removal operation time (*p* < 0.001) and the incidence of CXR bony spurs upon bar removal (*p* < 0.001) were both significantly lower in the DB group than in the NDB group (Table 1).

### 3.2. After Propensity Score Matching

A total of 48 PE patients were selected, with each group consisting of 24 patients after propensity score matching. There were no significant differences between the two groups in terms of age, gender, height, weight, BMI, HI, bar insertion operation time, complication rate, or bar removal hospital stay. However, the DB group had significantly shorter bar insertion hospital stay (*p* < 0.001) and bar removal operation time (*p* < 0.001) and a lower incidence of CXR bony spurs upon bar removal (*p* = 0.018) compared to the NDB group, even after propensity score matching (Table 2).

### 3.3. Risk Factors for Prolonged Bar Removal Time

Univariate and multivariate linear regression analyses were conducted to identify the potential risk factors associated with prolonged bar removal time. Univariate analysis revealed significant associations for several factors. NDB was significantly associated with a prolonged bar removal time (*p* < 0.001). Additionally, height (*p* = 0.035), weight (*p* = 0.041), and the presence of a CXR bony spur (*p* < 0.001) were identified as significant factors for prolonged bar removal time. A multivariate linear regression model indicated that NDB was a significant predictor of prolonged bar removal time (*p* < 0.001). The presence of a CXR bony spur also remained significantly associated with prolonged bar removal time (*p* < 0.001). Height and weight were not significant in the multivariate analysis. (Table 3)

## 4. Discussion

This study compared the clinical outcomes of MIRPE procedures between DB and NDB among PE patients. After propensity matching for baseline characteristics, the DB group exhibited shorter bar insertion hospital stays (in days), shorter bar removal operation times, and fewer incidences of CXR bony spurs compared to the NDB group. Multivariate analysis identified NDB use and the presence of CXR bony spurs as significant risk factors for prolonged bar removal operation time.

The fewer days spent in the hospital after bar insertion in DB the group could be attributed to more stable chest wall support through better force distribution, potentially reducing postoperative complications such as pain or displacement. Although MIRPE is widely recognized as an effective procedure, bar removal carries a risk of postoperative complications, including wound infection, pneumothorax, pleural effusion, cardiac perforation, and, in rare cases, even death [12]. While uncommon, these complications can prolong recovery, diminish patient satisfaction, and negatively impact overall quality of life [13]. The likelihood of such adverse events increases when technical challenges arise during removal, often occurring due to anatomical alterations, such as bony spurs forming around the implanted bar.

Bony spurs on a CXR appear as radiopaque structures, formed through granulation tissue development and subsequent osteocalcification, a process driven by inflammatory responses. The higher incidence of CXR bony spurs in the NDB group suggests that the NDB technique may exert unevenly distributed pressure on the chest wall, causing localized shear stress on the lateral stabilizers (hinge points), leading to bony spur formation. In contrast, the DB technique ensures better force distribution across the chest wall, as shown in our graphical abstract, thereby reducing the incidence of CXR bony spurs. Specifically, in addition to the two hinge points, the force acting on the chest wall (chest wall restoring force) is dispersed over multiple other points.

These bony spurs make bar removal surgeries more technically challenging. Surgeons often need to use orthopedic instruments, such as an osteotome or rongeur, to forcibly remove the bar, which increases operation time and the risk of wound complications that could result in greater postoperative pain [14]. Our findings are consistent with this observation, as the presence of CXR bony spurs was identified as a significant risk factor for prolonged removal operation time. In this study, postoperative complications following bar removal were observed in two patients, with both complications likely related to the excessive use of these orthopedic instruments during surgery causing tissue damage. To address these issues, we propose the DB technique as a strategy for possibly reducing the incidence of CXR bony spurs, shortening bar removal operation time, and consequently minimizing wound complications as well as perioperative pain and complications. Given that the primary patient population for this procedure consists of growing adolescents, minimizing surgical complications is critical [15]. In addition to improving patient outcomes, the DB procedure can also reduce the technical burden on surgeons during bar removal.

This study has a few limitations. First, as a retrospective study, there was potential for selection bias between the DB and NDB groups. To mitigate this, we conducted propensity score matching and performed analyses on the matched patients. Second, the final sample size was relatively small (n = 48), which may limit the generalizability of our findings. Third, while the DB technique appears to provide clinical benefits during both PE repair and bar removal, its long-term efficacy in maintaining chest wall correction post-removal remains uncertain. Lastly, bar maintenance time was consistent across patients, and it was not found to significantly influence the clinical outcomes of this study. Therefore, further prospective studies with larger sample sizes and long-term follow-ups are needed to validate these findings and further explore the clinical advantages of the DB technique.

While the MIRPE procedure offers significant advantages, including shorter recovery times and fewer complications, it may not be suitable for all patients. In particular, for patients with upper chest wall deformities, such as pectus arcuatum with complex deformity, open repair (a modified Ravitch operation) may provide a more effective correction [16]. However, for most younger patients with pliable chest walls, MIRPE remains a highly effective and minimally invasive option [17].

Given these considerations, our study provides valuable insights into the pros and cons of the DB and NDB techniques in MIRPE procedures gained through comparing their outcomes. By directly comparing both groups, we found that the DB technique offers significant advantages, including shorter bar removal operation times and fewer complications, making it the preferable option.

## Figures and Tables

**Figure 1 jcm-14-04217-f001:**
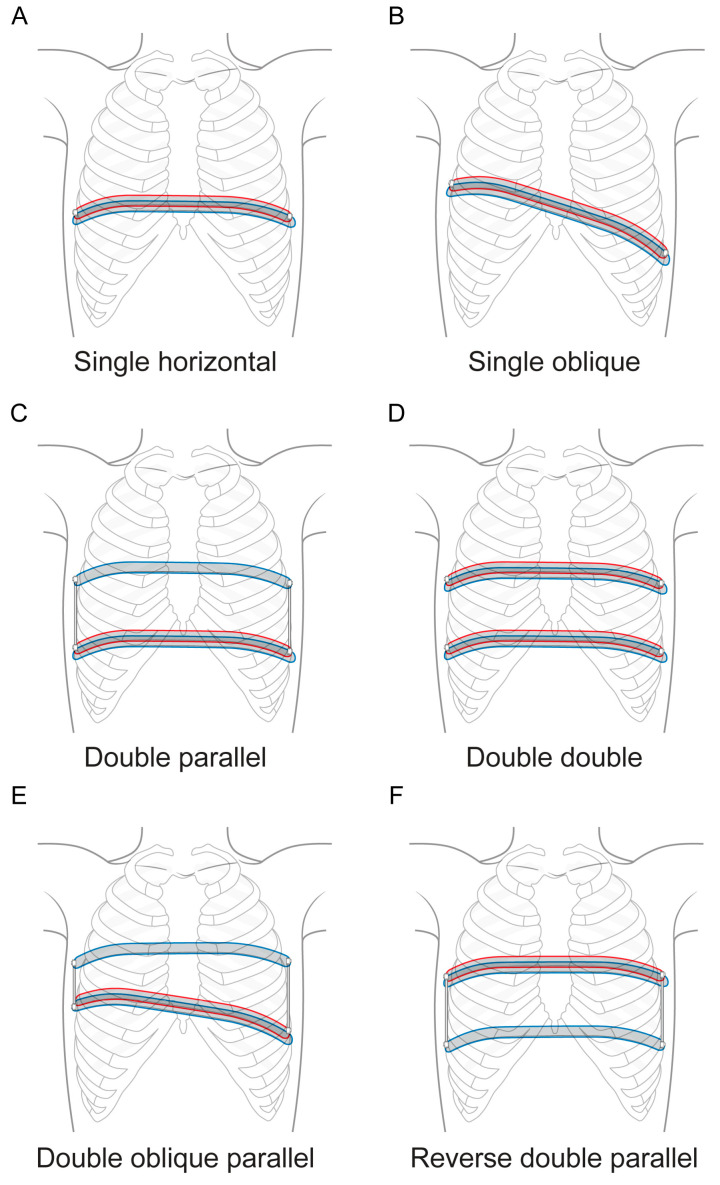
Various techniques of the double-bar method.

**Table 1 jcm-14-04217-t001:** Clinical characteristics of MIRPE patients before propensity score matching.

Variables	Double (n = 59)	Non-Double (n = 44)	*p*-Value
**Baseline characteristics**			
Sex			0.847
Female	13 (22.0)	9 (20.0)	
Male	46 (78.0)	35 (80.0)	
Age (years)	8.9 (7.4)	16.7 (5.4)	*** <0.001
Height (cm)	129.9 (27.9)	167.6 (14.3)	*** <0.001
Weight (kg)	29.3 (16.1)	51.7 (12.0)	*** <0.001
BMI (kg/m^2^)	16.6 (2.3)	18.5 (2.1)	*** <0.001
Preoperative HI	3.8 (1.0)	4.2 (1.0)	* 0.045
**Bar insertion variables**			
Operation time (minutes)	69.8 (17.6)	64.1 (16.4)	0.097
Hospital stay (days)	4.2 (0.7)	6.4 (1.4)	*** <0.001
Complications	0	5 (11.4)	* 0.012
**Bar removal variables**			
Operation time (minutes)	33.6 (6.7)	66.8 (26.3)	*** <0.001
Hospital stay (days)	2.1 (0.4)	2.4 (1.4)	0.204
Complications	0	2 (4.5)	0.098
CXR bony spurs	1 (1.7)	11 (25.0)	*** <0.001

Values are presented as n (%) or means (SD). BMI, body mass index; HI, haller index; CXR, chest X-ray. (* *p* < 0.05, *** *p* < 0.001).

**Table 2 jcm-14-04217-t002:** Clinical characteristics of MIRPE patients after propensity score matching.

Variables	Double (n = 24)	Non-Double (n = 24)	*p*-Value
**Baseline characteristics**			
Sex			>0.999
Female	6 (25.0)	6 (25.0)	
Male	18 (75.0)	18 (75.0)	
Age (years)	15.0 (8.4)	16.5 (6.5)	0.491
Height (cm)	156.8 (23.8)	163.9 (17.4)	0.239
Weight (kg)	44.4 (15.3)	49.5 (14.3)	0.238
BMI (kg/m^2^)	18.0 (2.6)	18.4 (2.4)	0.563
Preoperative HI	4.1 (1.0)	4.1 (0.9)	0.794
**Bar insertion variables**			
Operation time (minutes)	75.5 (23.2)	67.2 (18.7)	0.180
Hospital stay (days)	4.5 (0.8)	6.4 (1.2)	*** <0.001
Complications	0	3 (12.5)	0.234
**Bar removal variables**			
Operation time (minutes)	35.5 (7.1)	64.6 (27.9)	*** <0.001
Hospital stay (days)	2.1 (0.4)	2.4 (1.6)	0.408
Complications	0	1 (4.2)	0.312
CXR bony spurs	0	5 (20.8)	* 0.018

Values are presented as n (%) or means (SD). BMI, body mass index; HI, haller index; CXR, chest X-ray. (* *p* < 0.05, *** *p* < 0.001).

**Table 3 jcm-14-04217-t003:** Risk factors for prolonged bar removal time.

Variables	Univariate Linear Regression		Multivariate Linear Regression	
	β (SE)	*p*-Value	β (SE)	*p*-Value
NDB (vs. DB)	29.125 (5.884)	*** <0.001	21.477 (4.586)	*** <0.001
Height (cm)	0.367 (0.169)	* 0.035		
Weight (kg)	0.497 (0.236)	* 0.041		
CXR bony spur	50.833 (8.069)	*** <0.001	40.806 (7.046)	*** <0.001

NDB, non-double bar; DB, double bar; CXR, chest X-ray. (* *p* < 0.05, *** *p* < 0.001).

## Data Availability

The raw data supporting the conclusions of this article will be made available by the authors on request.

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
