# Peer review of "Comparison of the Outcomes of Double-Bar and Non-Double-Bar Techniques in the Minimally Invasive Repair of Pectus Excavatum: A Focus on Bar Removal and Complications"

_jcm, 2025, doi:10.3390/jcm14124217_

Round 1

Reviewer 1 Report

Comments and Suggestions for Authors

This is an interesting retrospective study about correction of pectus excavatum (PE) in children and adolescents using MIRPE. Most interesting is the fact, that double and non-double bar technique were compared. Especially complications and bar removal were analyzed comparing both techniques. It could be shown, that the outcome and removal of the double bar technique was better than in the non-double bar technique. The authors conclude, that the non-double bar technique led to more bony spurs, due to inequality of unevenly distributed pressure. These bony spurs may complicate the removal of the bar, which has been confirmed in literature.

There are nevertheless some points that should be better explained. Only after reading the study of Song et al. the technique of double compression was understandable. Therefore it would be better to explain it in your figures according to the figures in Song et al.

It is a pity that you cannot report on contentment of the patients with the procedure. It would be very helpful to know whether patients felt better after the procedure, particularly with regard to self-esteem and body image dissatisfaction.

You did describe several options for bar-insertion. How often did you use these 6 possibilities described in figure 1 and did you see differences in outcome?

Comments on the Quality of English Language

I am not a native English speaker, but some minor mistakes I could find (160: While uncommon, these complications can prolonged recovery e.g.)

Author Response

Comment 1: There are nevertheless some points that should be better explained. Only after reading the study of Song et al. the technique of double compression was understandable. Therefore it would be better to explain it in your figures according to the figures in Song et al.

Response 1: Thank you very much for your thoughtful comment. We truly appreciate your suggestion regarding the explanation of double compression techniques. We believe that the graphical representation you mentioned, which illustrates the points of action and magnitude of force with arrows, is already presented in our Graphical Abstract. To further clarify, we added the following in the discussion section which explains better force distribution of double-bar technique. We appreciate your feedback and hope this additional clarification provides a clearer understanding of double bar technique.

Line 170: In contrast, the DB technique ensures better force distribution across the chest wall, as shown in our Graphical abstract, thereby reducing the incidence of CXR bony spurs. Specifically, the force pushing on the chest wall (chest wall restoring force) is dispersed throughout multiple points, in addition to the two hinge points.

Comment 2: It is a pity that you cannot report on contentment of the patients with the procedure. It would be very helpful to know whether patients felt better after the procedure, particularly with regard to self-esteem and body image dissatisfaction.

Response 2: Thank you for your valuable comment. As this was a retrospective study, we unfortunately did not investigate self-esteem and body image dissatisfaction in detail. Most of the patients undergoing this surgery were young children, and it was difficult to assess these aspects. However, we plan to consider more objective scales in future studies to address this issue more thoroughly. We appreciate your feedback and will aim for a more comprehensive in the future.

Comment 3: You did describe several options for bar-insertion. How often did you use these 6 possibilities described in figure 1 and did you see differences in outcome?

Response 3: Thank you once again for your thoughtful comment. Regarding the six options for bar insertion described in Figure 1, we have reviewed the frequency of their use and found the following distribution: Type A – 33 patients, Type B – 4 patients, Type C – 16 patients, Type D – 2 patients, Type E – 3 patients, Type F – 1 patient. There were no significant statistical differences based on the method of bar-insertion. While we utilized different techniques depending on the specific needs of patients, there were no noticeable variations in complication rates or overall results between the different types of bar insertion. We greatly appreciate your valuable feedback, which allowed us to clarify this aspect in our study.

Comment 4: I am not a native English speaker, but some minor mistakes I could find (160: While uncommon, these complications can prolonged recovery e.g.)

Response 4: Thank you for pointing that out. We appreciate your attention to detail. The sentence in Line 160 has been revised to “While uncommon, these complications can prolong recovery e.g.” We have made this correction in the manuscript.

Reviewer 2 Report

Comments and Suggestions for Authors

The manuscript offers valuable insights into the comparative outcomes of double-bar versus non-double-bar techniques for MIRPE, particularly regarding bar removal. However, some key aspects require clarification to enhance the clinical applicability and scientific rigor of the findings.

  1. Bony Spurs Formation: The authors report a significantly lower incidence of bony spurs in the double-bar group but do not sufficiently explain why this configuration should mitigate spur development. It is widely recognized that bony spurs are often associated with localized mechanical stress and chronic inflammation, particularly near lateral stabilizers, which are more commonly employed in non-double-bar constructs. The potential role of stabilizers in spur formation should be explicitly addressed in the discussion.
  2. Reduced Hospital Stay and Complications: The manuscript suggests that patients undergoing the double-bar technique experience shorter postoperative stays, less pain, and fewer complications. However, given the added hardware and procedural complexity, these findings warrant further explanation. Are these outcomes linked to better force distribution, reduced movement of the bars, or avoidance of stabilizers? Clarifying the physiological or perioperative mechanisms would make the conclusions more convincing.
  3. Timing of Bar Removal: Surprisingly, the manuscript does not report the average interval between bar insertion and removal. Since bone remodeling and spur formation are time-dependent processes, and since longer indwelling times could increase removal difficulty, this information is essential for interpreting both the complication rates and the surgical time required for bar explantation.

Addressing these points would substantially strengthen the manuscript and help validate the claimed benefits of the double-bar technique.

Author Response

  1. Bony Spurs Formation: The authors report a significantly lower incidence of bony spurs in the double-bar group but do not sufficiently explain why this configuration should mitigate spur development. It is widely recognized that bony spurs are often associated with localized mechanical stress and chronic inflammation, particularly near lateral stabilizers, which are more commonly employed in non-double-bar constructs. The potential role of stabilizers in spur formation should be explicitly addressed in the discussion.

Response: Thank you for your valuable comment. In response to your feedback, we have added clarification in the Discussion section. Specifically, we now explain that non-double bar technique leads to localized mechanical shear stress around the lateral stabilizers (acting as hinge points), which contributes to bony spur formation. Also, we highlighted that double bar technique distributes force across multiple points, reducing this stress and lowering the incidence of CXR bony spurs. We hope this clarification provides a more explicit explanation of the role of lateral stabilizers and the advantages of double bar technique in reducing bony spur formation.

Line 167: The higher incidence of CXR bony spurs in the NDB group suggests that the NDB tech-nique may exert unevenly distributed pressure on the chest wall, causing localized shear stress on the lateral stabilizers (hinge points), leading to bony spur formation. In contrast, the DB technique ensures better force distribution across the chest wall, as shown in our Graphical abstract, thereby reducing the incidence of CXR bony spurs. Specifically, the force pushing on the chest wall (chest wall restoring force) is dispersed throughout multiple points, in addition to the two hinge points.

  1. Reduced Hospital Stay and Complications: The manuscript suggests that patients undergoing the double-bar technique experience shorter postoperative stays, less pain, and fewer complications. However, given the added hardware and procedural complexity, these findings warrant further explanation. Are these outcomes linked to better force distribution, reduced movement of the bars, or avoidance of stabilizers? Clarifying the physiological or perioperative mechanisms would make the conclusions more convincing.

Response: Thank you for your insightful comment. As noted in the first response, we have confirmed that the double bar technique, compared to the non-double bar technique, results in more even force distribution across the chest wall. This improved force distribution reduces the formation of bony spurs, which, in turn, contributes to a shorter bar removal surgery time, fewer postoperative complications and lesser postoperative pain. We further explained this relationship in the manuscript as follows.

Line 183: To address these issues, we propose the DB technique as a strategy to possibly reduce the incidence of CXR bony spurs, shorten bar removal operation time, and, consequently, minimize wound complications, as well as perioperative pain and complications during bar removal surgeries.

In addition, we recognized that the statement about shorter hospital days could be misleading, as it may be interpreted as referring to bar removal rather than bar insertion. To address this potential confusion, we have removed this statement and revised the text accordingly.

Line 153: After propensity matching for baseline characteristics, the DB group exhibited shorter bar removal operation times and lower incidences of CXR bony spurs compared to the NDB group.

We greatly appreciate your feedback, which helped improve the clarity of our study.

  1. Timing of Bar Removal: Surprisingly, the manuscript does not report the average interval between bar insertion and removal. Since bone remodeling and spur formation are time-dependent processes, and since longer indwelling times could increase removal difficulty, this information is essential for interpreting both the complication rates and the surgical time required for bar explantation.

Response: Thank you for your thoughtful comment. We acknowledge that the manuscript does not report the average interval between bar insertion and removal. In our practice, the bar is typically removed 2 years after insertion for patient under the age of 12, and after 3 years for patients above 12 years. However, the bar maintenance period is relatively consistent across patients, and we did not observe any significant impact on outcomes or complication rates due to slight variations in the removal timing. Factors such as the patient’s physical activity level and the shearing force on the chest wall could influence the timing of bar removal, but overall, we found that the maintenance period itself does not have a major effect on the outcomes. We appreciate your insightful suggestion and will consider addressing this issue in more detail in future studies.

Round 2

Reviewer 2 Report

Comments and Suggestions for Authors

1. Bony Spurs Formation

 Satisfactory

The authors provide a clear and plausible biomechanical rationale for the reduced incidence of bony spurs in the double-bar (DB) group. They appropriately attribute spur formation in the non-double-bar (NDB) group to localized mechanical stress around lateral stabilizers, and they emphasize that the DB technique distributes corrective forces more evenly across the chest wall. This addition enhances the scientific explanation and addresses the original concern effectively.

Verdict: Sufficiently addressed.

2. Reduced Hospital Stay and Complications

 Partially Satisfactory

The authors link reduced complications and postoperative pain in the DB group to the lower incidence of bony spurs and simpler bar removal procedures. This explanation is reasonable in the context of bar removal, but it does not fully explain why the initial postoperative hospital stay would be shorter in the DB group, given the technique involves more hardware.

Although the authors chose to remove the mention of shorter hospital stay (likely to avoid misinterpretation), a more robust discussion of early postoperative outcomes (e.g., pain control, stabilizer usage, mobility) would strengthen the argument.

Verdict: Acceptable, though slightly underdeveloped; no major concerns for publication, but further discussion would enhance clarity.

3. Timing of Bar Removal

 Partially Satisfactory

The authors acknowledge the omission and provide their institutional practice (2 years for patients <12; 3 years for ≥12). However, they did not include this information in the manuscript itself, nor did they discuss the potential confounding effect of bar duration on outcomes such as bony spur formation or surgical difficulty.

While their explanation (that bar maintenance time was relatively uniform) is reasonable, a brief mention of this variable in the Limitations section would increase transparency and scientific rigor.

Verdict: Acceptable, but should be explicitly stated in the manuscript.

Final Recommendation: Accept with Minor Revisions

The authors have addressed the key concerns adequately, with thoughtful and constructive revisions. Two minor additions would strengthen the manuscript and are recommended:

  1. Add a sentence in the Discussion or Limitations section noting that bar maintenance time was consistent across patients and not found to significantly influence outcomes.
  2. Expand slightly on the explanation of why the double-bar technique might lead to fewer complications or pain at the time of insertion, if supported by data or prior literature.

These are editorial rather than substantive revisions, and with their inclusion.

Author Response

1. Add a sentence in the Discussion or Limitations section noting that bar maintenance time was consistent across patients and not found to significantly influence outcomes.

Response: Thank you for your valuable suggestion. We have added a sentence in the Limitation in Discussion section noting that the bar maintenance time was relatively consistent across patients and was not found to significantly influence clinical outcomes in this study.

Line 198: Lastly, bar maintenance time was consistent across patients and was not found to significantly influence clinical outcomes in this study.

2. Expand slightly on the explanation of why the double-bar technique might lead to fewer complications or pain at the time of insertion, if supported by data or prior literature.

Response: Thank you for your insightful comment. While direct supporting literature is limited, we have reintroduced the shorter hospital stays after bar insertion in the DB group and added a brief explanation suggesting that more stable chest wall support through better force distribution might reduce early postoperative complications such as pain or bar displacement. This explanation has been included into the revised manuscript as follows.

Line 153: After propensity matching for baseline characteristics, the DB group exhibited shorter bar insertion hospital days, shorter bar removal operation times and lower incidences of CXR bony spurs compared to the NDB group.

Line 158: It is possible that the shorter bar insertion hospital days in DB the group could be attributed to more stable chest wall support through better force distribution, potentially reducing postoperative complications such as pain or displacement.

We sincerely appreciate your thoughtful feedback, which helped us improve the clarity and completeness of our manuscript.